# INFINITE-PARAMETER LARGE LANGUAGE MODEL

## ABSTRACT

In the standard transformer architecture, increasing model parameters leads to linear growth in computational cost and activation memory. To address this issue, we propose a novel Infinite Parameter Large Language Model (IP-LLM) architecture that decouples model size from computational cost and device memory.

Existing large language models are all fixed-parameter models, while human knowledge is infinite and expands daily. Finite parameters are inherently limited in their capacity to accommodate this boundless knowledge. Our IP-LLM architecture can potentially accommodate infinite knowledge, resolving this issue and laying the foundation for realizing a truly omniscient and omnipotent artificial general intelligence in the future.

Our architecture achieves performance comparable to MOE (Clark et al., 2022) while requiring significantly less memory.

## 1 INTRODUCTION

The emergence of Large Language Models (LLMs) has fundamentally transformed the landscape of natural language processing, demonstrating exceptional performance across a wide range of real-world applications (OpenAI, 2023; Chowdhery et al., 2023).

However, contemporary Large Language Models possess fixed parameter sizes, resulting in static knowledge upon completion of training. In contrast, human knowledge continuously expands, rendering models that require millions of dollars for training obsolete in a relatively short period, which is a considerable waste of resources.

Updating knowledge in fixed-parameter Large Language Models poses significant challenges due to catastrophic forgetting (De Lange et al., 2021), often resulting in a decline in their original general capabilities (Cheng et al., 2023; Dong et al., 2023). This necessitates a methodology that enables large models to retain previously acquired knowledge while simultaneously facilitating the continuous learning of new information.

Scaling laws for neural language models show the power of scaling (Kaplan et al., 2020; Hoffmann et al., 2022): increasing the number of parameters, amount of training data, or the computational budget has proven to be a reliable way to improve model performance. However, there is a linear relationship between computational footprint, as measured by FLOPs and device memory consumption, and parameter count.

To address these challenges, we propose a novel Infinite-Parameter Large Language Model (IP-LLM) architecture. Unlike traditional fixed-parameter models, IP-LLM cleverly decouples the relationship between the model parameter scale and the linear cost of computation and device memory. Our core strategy is to divide the model parameters into multiple independent experts, with each expert responsible for storing knowledge in a specific category. During inference, IP-LLM only loads the experts relevant to the current task, significantly reducing computation costs and memory consumption.

Specifically, IP-LLM employs a routing-based dynamic parameter selection mechanism. This mechanism first determines the category of the input text based on its content and context. Then, IP-LLM loads only the expert parameters corresponding to the relevant category for computation, while experts unrelated to the current task remain unloaded. This dynamic parameter loading approach enables the model to effectively handle an infinite amount of knowledge without being constrained by memory and computational resources. This is conceptually similar to the Mixture of Experts

(MoE) approach, but our method is more efficient as it leverages the large model's language understanding capabilities to perform routing, rather than relying on only a subset of parameters as in MoE, thereby improving routing accuracy. Additionally, we adopt a staged pretraining strategy, first pretraining basic language knowledge, then training domain-specific knowledge, and reducing the total parameter count by sharing parameters.

Through this design, IP-LLM enables the realization of an "infinite-parameter" large language model, while requiring only minimal inference memory and computational cost.

The model architecture is shown in Figure 1 . To validate the feasibility of this architecture, we partitioned the data into 22 categories and designed a 24B-parameter model. The model comprises 24.5 billion parameters, of which 7.2 billion are dedicated to the base component, 0.7 billion to the routing component, and the remaining 16.6 billion are distributed across 22 distinct categories. During inference, only the 7.2B base, 0.75B router, and the parameters for a single data category (0.75B) are loaded into memory, totaling 8.7B.

This represents a 65% reduction in both inference memory and computational cost compared to a fixed 24.5B parameter model.

Compared to an MoE model with 24.5B parameters and 22 experts, the inference memory consumption is significantly reduced.

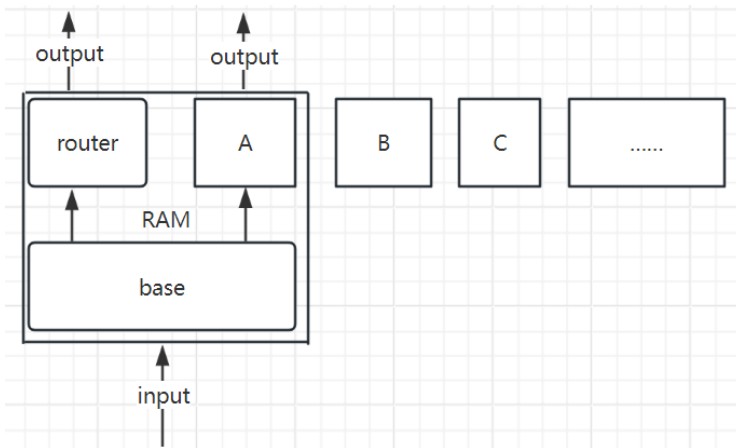

Figure 1: Parameters A, B, C, and D store knowledge for different categories. When reasoning about Category A problems, only parameter A needs to be loaded into memory, eliminating the need to load all parameters.

This paper makes the following contributions:

- Inspired by the routing mechanism of MoE, this paper proposes a novel approach that leverages all model parameters for routing, instead of a subset, significantly enhancing routing accuracy.

  The method involves first utilizing prompt words to enable the model to predict and classify the input text into a specific domain, followed by inference using parameters specialized for that domain.

- We proposes a segmented pretraining framework, separating the pretraining process into two phases. The first phase emphasizes the acquisition of foundational linguistic knowledge, including lexical, grammatical, and syntactic elements, as well as basic world knowledge. The second phase then focuses on learning knowledge built upon this linguistic foundation.

- We propose a novel infinite-parameter large language model capable of lifelong learning without catastrophic forgetting, by strategically training new knowledge onto fresh parameters.

- This innovation results in a drastic reduction in both training cost and inference memory consumption for the large language model. We observe a significant decrease in training cost, while inference memory consumption is lowered by approximately 65%.

## 2 RELATED WORK

### 2.1 CATASTROPHIC FORGETTING

Preventing catastrophic forgetting during training remains a classic challenge in deep learning. Since 2018, numerous studies have explored strategies to mitigate this issue for Large Language Models (LLMs). For instance, (Yang et al., 2024) introduced a self-distillation method that addresses the distribution gap between task datasets and LLMs, effectively reducing catastrophic forgetting while maintaining general capabilities.

In empirical investigation, (Luo et al., 2024) revealed that as the scale of Large Language Models (LLMs) increases during continual instruction tuning, the incidence of catastrophic forgetting becomes more pronounced. They suggested that employing general instruction tuning could alleviate this challenge.

(Hsieh et al., 2023) introduced a novel approach for training smaller models, demonstrating that these models can outperform LLMs with less training data by utilizing rationales extracted from LLMs as additional supervision. Furthermore, (Wang et al., 2023b) demonstrated O-LoRA, a method designed to mitigate catastrophic forgetting by learning new tasks in orthogonal subspaces, thereby minimizing interference with previously acquired knowledge.

### 2.2 MIXTURE OF EXPERTS

Several recent works (Shazeer et al., 2017; Lepikhin et al., 2020; Fedus et al., 2022; Zhou et al., 2022) have adopted the Mixture-of-Experts (MoE) architecture to decouple computational cost from parameter count.

(Geva et al., 2021; Dai et al., 2022) argue that feedforward (FFW) layers store factual knowledge .These layers constitute approximately two-thirds of the total parameters in a transformer architecture. The Mixture-of-Experts (MOE) architecture deviates from the traditional single dense feedforward network (FFW) by utilizing a set of sparsely activated expert modules, frequently implemented as FFWs.

(Clark et al., 2022) investigated the scaling properties of MoE language models, demonstrating that increasing the number of experts can effectively enhance performance without incurring additional inference costs. However, their experiments revealed that the efficiency gains offered by MoEs plateau after reaching a particular model size.

More recently, (Krajewski et al., 2024) identified that this plateauing phenomenon was a consequence of using a fixed number of training tokens. Their findings demonstrate that when the number of training tokens is optimized for computational efficiency, MoEs consistently outperform dense models in terms of FLOPs (floating-point operations) per parameter. Furthermore, they introduced granularity, the number of active experts, as a novel scaling dimension. Their empirical studies revealed that employing higher granularity leads to improved performance.

### 2.3 PROGRESSIVE LEARNING

The concept of progressive training has attracted significant attention for its potential to expedite the training of large-scale models in both computer vision (Zhang et al., 2023) and natural language processing (NLP) (Yao et al., 2023; Li et al., 2023).

A stacking technique that successively doubles model depth was proposed by (Gong et al., 2019). Enhancing this approach, CompoundGrow (Gu et al., 2020) integrates Feed-Forward Network expansion into its scheduling framework. Moreover, (Shen et al., 2022) introduced a method that supports the expansion of hidden sizes in a staged manner. Notably, both Bert2BERT (Chen et al., 2021) and LiGO (Wang et al., 2023a) are designed to accommodate all dimensions of growth.

## 2.4 LIFELONG LEARNING / CONTINUAL LEARNING USING MOE

(Chen et al., 2023) proposed using MoE (Mixture of Experts) for lifelong learning in large language models (LLMs), referred to as "Lifelong MoE." However, it has limitations: it can only mitigate catastrophic forgetting but cannot prevent it entirely. Moreover, adding new experts may lead to performance degradation in routing for old tasks. The innovations of "IP-LLM" lie in the following aspects:

**Infinite Parameter Model Architecture** : IP-LLM introduces the concept of infinite parameters by grouping parameters and using on-demand loading. During inference, only the required parameters are loaded, theoretically surpassing the limitations of model size. While Lifelong MoE also expands the model by adding experts, its total parameters remain finite.

**On-Demand Parameter Loading** : A core innovation of IP-LLM is its on-demand parameter loading mechanism. The model's parameters are divided into multiple groups, with each group corresponding to a specific type of knowledge or domain. During inference, only the parameter groups relevant to the current task are loaded, significantly reducing memory usage. Although Lifelong MoE also utilizes a subset of experts during inference, it requires all experts to be loaded into GPU memory, limiting parameter scalability.

**Improvements in Pretraining** : IP-LLM introduces a staged pretraining framework. It first learns fundamental language knowledge (e.g., vocabulary, grammar), then trains parameters on different data categories separately, and finally integrates them. This approach helps reduce the parameter size of individual experts.

**Avoiding catastrophic forgetting** : Adding new experts can be done without modifying the parameters of existing experts, thereby preventing catastrophic forgetting.

## 2.5 OTHER RESEARCH

The Branch-Train-Merge (BTM) (Li et al., 2022) method decomposes large language models (LLMs) into multiple independent expert models, which are trained in parallel and then merged for inference. The main goal is to accelerate training. There are two merging approaches: **Ensemble**, where all models are run and their results are weighted and averaged, leading to high inference costs; and **Parameter Averaging**, where the parameters of all expert models (ELMs) are weighted and averaged before running, reducing inference costs but leading to worse performance.

The downside is that the merging process often leads to performance degradation. Using different weights for different ELM models can alleviate this issue, but selecting the optimal weights still depends on the specific task and dataset, which is cumbersome.

Branch-Train-MiX (BTX) (Sukhbaatar et al., 2024) is an improvement of BTM, where the merging process is changed to turn ELMs into experts in an MoE model, adding a token-level routing mechanism and then fine-tuning. However, the drawback is that averaging parameters across different layers of the ELMs can lead to performance degradation, making it unstable. After merging into MoE, the parameter size increases significantly, requiring much more memory.

BTM, BTX, and IP-LLM differ significantly, mainly in the following aspects:

**Difference in Objectives** : The primary aim of BTM and BTX is to improve training speed. In contrast, the main goal of IP-LLM is to achieve an infinitely large model, dynamically adding parameters, avoiding catastrophic forgetting when learning new domain knowledge, and requiring minimal retraining.

**Architectural Differences** : In BTM and BTX, the parameters of different ELM models are independent, with no shared parameters, leading to inefficient use of parameters.

In IP-LLM, different experts share most parameters, with common parameters for general language skills and basic knowledge, and only a small set of domain-specific parameters differing, making it highly efficient.

After training, BTM and BTX require merging, which can cause unpredictable performance losses. In contrast, IP-LLM does not require merging and avoids such performance loss.

**Routing Differences** : IP-LLM uses the language understanding ability of large language models for routing, rather than relying on just a few simple layers of neural networks, resulting in much higher accuracy.

## 3 TASK DEFINITION

### 3.1 THE TRANSFORMER BLOCK

The transformer block yields an output $y$ for an input $x$, as articulated by the following equations. It is structured with a multi-head self-attention (MHSA) mechanism, followed by a position-wise feed-forward network (FFN) that integrates residual connections and utilizes a Swish-Gated Linear Unit (SwiGLU) operation.

$$
\begin{aligned}
x' &= x + \text{MHSA}(\text{RMSNorm}(x)) \\
y &= x' + \text{FFN}(\text{RMSNorm}(x'))
\end{aligned}
\tag{1}
$$

The multi-head self-attention (MHSA) operation constitutes a critical component of the transformer architecture and is defined as follows. The input $x$ has a dimension of $n \times d$, with $n$ representing the sequence length and $d$ denoting the hidden size. The output $y$ retains the same dimensionality as the input $x$.

$$
\text{MHSA}(Q, K, V) = \text{Concat}(\text{head}_1, \ldots, \text{head}_h)W^O
\tag{2}
$$

where $Q$, $K$, and $V$ represent the query, key, and value matrices, respectively, and $W^O$ denotes the output weight matrix, which is applied without bias.Each head is computed as follows:

$$
\begin{aligned}
\text{head}_i &= \text{Attention}(xW_i^Q, xW_i^K, xW_i^V) \\
\text{Attention}(Q_i, K_i, V_i) &= \text{Softmax}\left(\frac{Q_i K_i^T}{\sqrt{d_k}}\right) V_i
\end{aligned}
\tag{3}
$$

where the corresponding weight matrices for the $i$-th head are denoted as $W_i^Q$, $W_i^K$, and $W_i^V$ .

The SwiGLU activation function, which is defined as follows, is utilized in the FFN block of the transformer block:

$$
\begin{aligned}
\text{SiLU}(x) &= x \otimes \sigma(x) \\
\text{SwiGLU}(x, W, V) &= \text{SiLU}(xW) \otimes (xV) \\
\text{FFN}(x) &= \text{SwiGLU}(x, W_1, W_2)W_3
\end{aligned}
\tag{4}
$$

where $W_1$, $W_2$, and $W_3$ are the weight matrices without bias,$\otimes$ denotes element-wise multiplication .

### 3.2 ROUTING MECHANISM

Our model utilizes a mechanism for selective parameter loading during inference, enabling successful reasoning even under memory constraints. Only a small subset of parameters is required to be retained in memory.

We define a given model $f$ that comprises three sets of specified models $f_{base}$,$f_{router}$, many expert models( $f_A$,$f_B$,$f_C$ ,$f_D$ ,$f_E$ ...) and a set of input-output pairs $(x, y)$. we can define this process as:

$$x' = f_{base}(x_i) \tag{5}$$

$f_{base}$ represents the inference process of the parameters in the base part. The input x is subject to parsing via the $f_{base}$

$$R = f_{router}(x') \tag{6}$$

$f_{router}$ represents the inference process of the parameters in the routing part. After passing through the $f_{router}$, we obtain R, which signifies the category to which the input belongs.

$$f(x_i) = \begin{cases} f_A(x') & \text{if } R = TokenA \\ f_B(x') & \text{if } R = TokenB \\ f_C(x') & \text{if } R = TokenC \\ f_D(x') & \text{if } R = TokenD \\ f_E(x') & \text{if } R = TokenE \\ ... \\ x' & \text{if } R \in other \end{cases} \tag{7}$$

$f_A$ represents the inference process of the parameters that encode domain knowledge from A. $f_B$ and $f_C$ follow the same pattern. Based on the determined category, it select corresponding parameters for inference.

## 3.3 KNOWLEDGE UPDATING

For traditional fixed-parameter models, if you need to add knowledge in a specific category, it inevitably causes a decrease in the performance of knowledge in other categories . However, our model can enhance the performance of a specific category without affecting the performance of other categories. Because in our model, knowledge from different categories is stored in different parameters, they don't affect each other. When there is a need to update knowledge in a specific category, such as category B, only $f_B$ needs to be adjusted. Parameters for other categories do not need to be changed, and it won't affect the performance of other categories. If the knowledge in category b increases significantly, additional parameters can be added to $f_B$, and $f_B$ can be retrained. Parameters for other categories do not need to be modified.

## 3.4 LEARNING KNOWLEDGE IN A NEW CATEGORY

The routing token set is open. When we need to generalize to unseen categories, we only need to add a set of parameters $f_Y$ and a routing token TokenY. Then, train this set of parameters with the new knowledge and retrain the router using the labels of both the new and old knowledge.

$$R = f_{router}^{new}(x') \tag{8}$$

$$f(x_i) = \begin{cases} f_A(x') & \text{if } R = TokenA \\ f_B(x') & \text{if } R = TokenB \\ f_C(x') & \text{if } R = TokenC \\ f_D(x') & \text{if } R = TokenD \\ f_E(x') & \text{if } R = TokenE \\ ... \\ f_Y(x') & \text{if } R = TokenY \\ ... \\ x' & \text{if } R \in other \end{cases} \tag{9}$$

## 3.5 GENERALIZE TO INFINITELY MANY CATEGORIES

When a large number of categories are added, if the routing accuracy decreases, it can be ensured by appropriately increasing the parameter capacity of $f_{router}$ and retraining it.

## 4 TRAINING STRATEGY

**Training data**  The data is filtered from open-source datasets, including The Pile, Skypile, RefinedWeb, RedPajama and common crawl, as well as some synthetic data.Approximately 1 trillion tokens in total.

The dataset is comprised of two distinct components. The first part focuses on training a base model, emphasizing foundational linguistic knowledge including vocabulary, grammar, syntax, and basic world knowledge. The second part consists of domain-specific knowledge, used to train the router and specialized parameters for each domain.

**base model**  As a first step,in consideration of computational resource constraints, we employ the Qwen1.5-beta-7B-Chat  (Bai et al., 2023) model ,a pre-trained language model with strong performance in various tasks,as the base model.

**Expert layer**  Subsequently, we replicate the last transformer layer of the base model four times and append these four transformer layers to the end of the base model's last transformer layer. We then mix domain-specific data and general data in a defined proportion to train the parameters of these four layers, thereby facilitating the acquisition of specialized knowledge, while the remaining parameters are kept frozen.

**routing layer**  After training, the new four transformer layers replace the previous four layers, and the process is repeated for other domains.

Finally, we add four transformer layers after the last transformer layer of the base model to serve as a router. This router is trained using a dataset composed of all domain-specific data, where each data point is labeled with its corresponding domain.

## 5 EXPERIMENTS

### 5.1 DATASETS

For evaluation, we use the few-shot performance on multiple benchmarks that test different skills:

**Popular aggregated results:**MMLU (Hendrycks et al., 2020) (5-shot) and C-Eval  (Huang et al., 2024) (5-shot)

**Math:** GSM8K (Cobbe et al., 2021) (8-shot) with maj@8 and MATH (Hendrycks et al., 2021) (4-shot) with maj@4

The evaluation results (as shown in Figure 2) indicate that our trained model demonstrates performance comparable to the dense model. However, both the training cost and inference cost are significantly reduced.

### 5.2 COMPARISON WITH MOE

IP-LLM and MoE share the concept of expert networks but differ in several key aspects:

ROUTING MECHANISM:

IP-LLM uses the base model for general language understanding in conjunction with routing parameters, resulting in a significantly larger total parameter size compared to MoE, which typically uses only a small subset of parameters. This leads to higher routing accuracy in IP-LLM and more efficient utilization of the model's existing parameters.MoE performs routing at every layer of the

| Model | MMLU | C-Eval | GSM8K | MATH |
|---|---|---|---|---|
| GPT-4 | 86.4 | 69.9 | 92.0 | 45.8 |
| Llama2-7B | 46.8 | 32.5 | 16.7 | 3.3 |
| Llama2-13B | 55.0 | 41.4 | 29.6 | 5.0 |
| Llama2-34B | 62.6 | 44.6 | 42.2 | 6.2 |
| Llama2-70B | 69.8 | 50.1 | 54.4 | 10.6 |
| Mistral-7B | 64.1 | 47.4 | 47.5 | 11.3 |
| Mixtral-8x7B | 70.6 | 52.1 | 74.4 | 28.4 |
| Qwen1.5-7B | 61.0 | 74.1 | 62.5 | 20.3 |
| IPLLM-24B | 75.2 | 86.7 | 80.3 | 35.5 |
| Qwen1.5-32B | 73.4 | 83.5 | 77.4 | 36.1 |
| Qwen1.5-72B | 77.5 | 84.1 | 79.5 | 34.1 |

Figure 2: Comparison.

transforms, while IP-LLM only performs routing once during sentence generation, until the prediction is complete.

For example, the routing mechanism in IP-LLM 24B utilizes a 7.2B base model and a 0.7B router, amounting to a total of 7.9B parameters, which accounts for approximately 33% of the total 24B parameters. In contrast, the MOE architecture only adds a few fully connected layers before the expert layers as its router. Based on our calculations, the routing parameter size of Mixtral 8x7B is less than 1B, significantly smaller than that of our model. Therefore, in theory, the routing precision of our IP-LLM model is far superior to that of MOE.

KNOWLEDGE UPDATING:

IP-LLM enables incremental learning of new knowledge by adding new parameter blocks, requiring only the training of new parameters and the router. In contrast, MoE generally requires retraining the entire model. This makes IP-LLM more adaptable to evolving environments and knowledge.

MEMORY EFFICIENCY:

During inference, MoE requires loading all parameters into memory, while IP-LLM only loads a subset of parameters, making it more memory-efficient than MoE.

COMPUTATIONAL EFFICIENCY

Both MoE and IP-LLM involve only a subset of parameters in computation. The actual computational efficiency depends on the number of experts and the proportion of expert parameters to the total parameters. When the total amount of expert parameters and the number of experts are the same, MoE requires activating at least two experts and averaging their outputs during computation, which leads to computational waste. In contrast, IP-LLM only needs to activate a single expert, making its computational efficiency higher.

### 5.3 ABLATION STUDY

#### 5.3.1 ROUTING STRATEGY

**Routing at each Transformer layer**   : In MOE, the routing strategy selects a route at each Transformer layer, which makes it impossible to predict which expert should be loaded before inference. As a result, all experts need to be loaded into memory. Given the small memory size of current devices, this clearly cannot support an infinitely large model.

**Routing at each token prediction**   : Before inference, it is possible to predict which expert to load. This allows only the specified expert to be loaded into memory, without the need to load all parameters, enabling models with parameters far larger than the available memory. However, the performance is significantly slowed down due to switching experts for each token.

**Routing at the start of each inference task**   : Suppose a task requires predicting 100 tokens. Routing is only performed once before the first token, after which the expert is switched, and the same expert is used for predicting the following 100 tokens. This reduces the routing overhead and expert switching costs to a minimum, making it suitable for infinitely large models.

#### 5.3.2 NUMBER OF LAYERS

The number of layers for each category's parameters depends on the amount of knowledge contained in that category. The more categories there are, the less knowledge each category contains, and thus, fewer parameters are required for each category.

#### 5.3.3 BASE MODEL

The more knowledge the base model contains, the fewer parameters the experts need, but the inference cost increases. In the extreme case, where the base model contains all knowledge, the experts' parameters are zero, resulting in the highest inference cost.

The base model should aim to contain shared knowledge required across all domains. Reducing the amount of knowledge in the base model increases the number of parameters required by the experts, but decreases the inference cost.

## 6 CONCLUSION

In this paper, we introduce a novel architecture for large language models that offers significant advantages in terms of reduced device memory requirements for both training and inference, while also enabling the model to learn new knowledge without catastrophic forgetting.

## 7 LIMITATIONS

In this work, we did not train the base model from scratch due to computational constraints. Training the base model from scratch might further enhance performance. We will address the issue of multi-domain knowledge fusion in a subsequent paper.

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
