# OpenReview forum: "Infinite-parameter Large Language Model"
_ICLR.cc/2025/Conference — Submitted to ICLR 2025_

### Official Review · Reviewer_NVv4 · 2024-10-31

**Soundness:** 2
**Presentation:** 1
**Contribution:** 2
**Rating:** 3
**Confidence:** 3

**Summary:**

The paper presents IP-LLM inspired from MoE which use routing mechanism to enable continual learning of multi-domain tasks and memory efficient inference. The authors use segmented pre-training strategy to train a base block to acquire general linguistic skills and then train router and domain-specific modules.
The authors evaluate IP-LLM's performance on 4 different tasks.

**Strengths:**

The authors propose a novel pre-training strategy to parse the general linguistic comprehension skills in the base model and later train individual experts on top on different domains.

**Weaknesses:**

1. Evaluation

- I think the biggest issue in current manuscript is evaluation of the model. While the current evaluation only includes monolithic architectures without MoE strategy, wouldn't it be fairer to include the models using MoE's in terms of both performance and memory/compute efficiency?

- The authors claim in the list of contributions that the new approach allows higher routing accuracy but I cannot find the explicit result for this.

- Also the authors claim the memory and training efficiency but having a explicit numerical comparison and what exact 'training cost' is meant here.

2. Related works on lifelong learning / continual learning using MOE

- I believe there are already few literatures on lifelong learning of LLM using MoE e.g Chen et al (2023) https://arxiv.org/pdf/2305.12281. I suggest authors to incorporate more relevant literatures and what is the novelty of their method.

3. Paper presentation should be improved

- There are many typos and inconsistent citing notations which makes readability very low. For example, I found Section 2 related work very hard to parse the included citations (spacing, parentheses etc). I highly recommend authors to do careful proofreading of the entire manuscript.

- Section 4 training strategy can be improved with adding a schema for better delivery.

**Questions:**

I believe my comments about the possible improvements and weaknesses incorporate my questions.

---

> ### Author Response · Authors · 2024-11-16
>
> Thank you sincerely for taking the time to review our work and provide such valuable feedback. We greatly appreciate your insights and are thoroughly addressing the points raised in the Weaknesses section, integrating them into the revised version of the paper.
>
> # 1.Weaknesses1.1
>
> **Comparison with MoE:**
> IP-LLM and MoE share the concept of expert networks but differ in several key aspects:
>
> - **Routing Mechanism:**
>   IP-LLM uses the base model for general language understanding in conjunction with routing parameters, resulting in a significantly larger total parameter size compared to MoE, which typically uses only a small subset of parameters. This leads to higher routing accuracy in IP-LLM and more efficient utilization of the model's existing parameters.
>
> - **Knowledge Updating:**
>   IP-LLM enables incremental learning of new knowledge by adding new parameter blocks, requiring only the training of new parameters and the router. In contrast, MoE generally requires retraining the entire model. This makes IP-LLM more adaptable to evolving environments and knowledge.
>
> - **Memory Efficiency:**
>   During inference, MoE requires loading all parameters into memory, while IP-LLM only loads a subset of parameters, making it more memory-efficient than MoE.
>
> - **Computational Efficiency:**
>   Both MoE and IP-LLM involve only a subset of parameters in computation. The actual computational efficiency depends on the number of experts and the proportion of expert parameters to the total parameters. When the total amount of expert parameters and the number of experts are the same, MoE requires activating at least two experts and averaging their outputs during computation, which leads to computational waste. In contrast, IP-LLM only needs to activate a single expert, making its computational efficiency higher.
>
>
>
> # 2.Weaknesses1.2
>
> **Routing Precision:**
>
> The routing mechanism in IP-LLM 24B utilizes a 7.2B base model and a 0.7B router, amounting to a total of 7.9B parameters, which accounts for approximately 33% of the total 24B parameters. In contrast, the MOE architecture only adds a few fully connected layers before the expert layers as its router. Based on our calculations, the routing parameter size of Mixtral 8x7B is merely 0.001B, significantly smaller than that of our model. Therefore, in theory, the routing precision of our IP-LLM model is far superior to that of MOE.
>
> # 3.Weaknesses1.3
>
> - **Memory Efficiency:**
>   During inference, MoE requires loading all parameters into memory, while IP-LLM only loads a subset of parameters, making it more memory-efficient than MoE.
> - **Training Efficiency:**
>   IP-LLM achieves incremental learning of new knowledge by adding new parameter blocks, requiring only the training of new parameters and the router, whereas MoE generally requires retraining the entire model. IP-LLM offers higher training efficiency when learning new knowledge.
>
> # 4.Weaknesses2
>
> Chen et al. (2023)  https://arxiv.org/pdf/2305.12281 proposed using MoE (Mixture of Experts) for lifelong learning in large language models (LLMs), referred to as "Lifelong MoE." However, it has limitations: it can only mitigate catastrophic forgetting but cannot prevent it entirely. Moreover, adding new experts may lead to performance degradation in routing for old tasks. The innovations of "IP-LLM" lie in the following aspects:
>
> - **Infinite Parameter Model Architecture:** IP-LLM introduces the concept of **infinite parameters** by grouping parameters and using on-demand loading. During inference, only the required parameters are loaded, theoretically surpassing the limitations of model size. While Lifelong MoE also expands the model by adding experts, its total parameters remain finite.
> - **On-Demand Parameter Loading:** A core innovation of IP-LLM is its **on-demand parameter loading** mechanism. The model's parameters are divided into multiple groups, with each group corresponding to a specific type of knowledge or domain. During inference, only the parameter groups relevant to the current task are loaded, significantly reducing memory usage. Although Lifelong MoE also utilizes a subset of experts during inference, it requires all experts to be loaded into GPU memory, limiting parameter scalability.
> - **Improvements in Pretraining:** IP-LLM introduces a **staged pretraining** framework. It first learns fundamental language knowledge (e.g., vocabulary, grammar), then trains parameters on different data categories separately, and finally integrates them. This approach helps reduce the parameter size of individual experts.
> - **Avoiding catastrophic forgetting:** Adding new experts can be done without modifying the parameters of existing experts, thereby preventing catastrophic forgetting.
> # 5.Weaknesses3
> We will address these issues in the revised version.
>
>
> We hope our response addresses your concerns, but if there are any issues or areas we missed, please feel free to point them out.

---

> > ### Author Response · Authors · 2024-11-26
> >
> > We have resolved all the issues in the weaknesses section and updated   section  5.2 and  2.4 in the revised version.

---

> ### Comment · Reviewer_NVv4 · 2024-11-29
>
> Thanks for the response,
> but I believe the current manuscript is not yet convincing and I'll keep my current score. Lots of details of the experiments are missing, it is still not clear what is novel compare to previous works and the manuscript is not well structured and poorly written.
> I encourage authors to improve the delivery of the idea and evaluation.

---

### Official Review · Reviewer_K88R · 2024-11-04

**Soundness:** 2
**Presentation:** 2
**Contribution:** 2
**Rating:** 3
**Confidence:** 5

**Summary:**

The paper proposes an "Infinite-Parameter Large Language Model" with the idea to accommodate the increasing amount of information generated in the world that it hypothesizes models with a fixed number of parameters will eventually not be able to contain. The implementation involves training the model as a Mixture of Experts.

**Strengths:**

The paper tackles an important question.

**Weaknesses:**

- The paper does not make a distinction between the proposed approach from MoE training. And if there is indeed a difference, please include the MoE baseline.
- **Major issue**: The paper compares their model trained on downstream tasks with other pre-trained models, zero-shot on the downstream tasks. Therefore, it is not making an apples-to-apples comparison
- It would be important to add in a couple of baselines to showcase the benefit of the proposed method over others
  - A single model trained on all the data that the base, router, and individual experts are trained on
  - MoE baseline trained on the data used to train the IP-LM.
- Why are there no entries in the table for some models on C-Eval?
- There are not many details provided on training, model architecture, and dataset. Unclear what data is additionally used to train the base model. What is the architecture of the model?
- The writing in the paper is clear but not precise. For example, the abstract or intro does not tell you anything concrete about what the paper builds, it only goes so far as to specify the problem and motivate it.

**Questions:**

Please take a look at the section on weaknesses.

---

> ### Author Response · Authors · 2024-11-17
>
> # 1.Weaknesses1
>
> **Comparison with MoE:**
> IP-LLM and MoE share the concept of expert networks but differ in several key aspects:
>
> - **Routing Mechanism:**
>   IP-LLM uses the base model for general language understanding in conjunction with routing parameters, resulting in a significantly larger total parameter size compared to MoE, which typically uses only a small subset of parameters. This leads to higher routing accuracy in IP-LLM and more efficient utilization of the model's existing parameters.  MoE performs routing at every layer of the transforms, while IP-LLM only performs routing once during sentence generation, until the prediction is complete.
>
> - **Knowledge Updating:**
>   IP-LLM enables incremental learning of new knowledge by adding new parameter blocks, requiring only the training of new parameters and the router. In contrast, MoE generally requires retraining the entire model. This makes IP-LLM more adaptable to evolving environments and knowledge.
>
> - **Memory Efficiency:**
>   During inference, MoE requires loading all parameters into memory, while IP-LLM only loads a subset of parameters, making it more memory-efficient than MoE.
>
> - **Computational Efficiency:**
>   Both MoE and IP-LLM involve only a subset of parameters in computation. The actual computational efficiency depends on the number of experts and the proportion of expert parameters to the total parameters. When the total amount of expert parameters and the number of experts are the same, MoE requires activating at least two experts and averaging their outputs during computation, which leads to computational waste. In contrast, IP-LLM only needs to activate a single expert, making its computational efficiency higher.
>
> # 2.Weaknesses2
>
> IP-LLM is also a pre-trained model and belongs to the same category of models.
>
> # 3.Weaknesses3
>
> We fully agree with your point of view, but due to computational limitations, we have not conducted such experiments yet. Theoretically, because the knowledge gap between different categories is large, mixing them during training could lead to interference and result in loss. Therefore, we speculate that if data classification is done well enough, under the same data, parameter count, and computational resources, having IP-LLM separate the data and train each expert could yield better results than a single model.
>
>
> In the early and mid-stages of training, the MoE model suffers from inaccurate routing, which causes a large amount of data to be routed to the wrong experts for training, leading to significant waste of data and computational resources, and even performance degradation. In contrast, IP-LLM has 100% accurate routing throughout the entire training process. Therefore, we predict that, with the same data, parameters, and computational resources, IP-LLM will outperform MoE in training effectiveness.
> # 4.Weaknesses4
>
> It has been corrected in the revised version.
>
> # 5.Weaknesses5
>
> The model architecture is very simple. It adds 4 interchangeable layers of parameters after the last Transformer layer of Qwen1.5-7B. When these 4 layers are routing parameters, routing decisions can be made. When these 4 layers contain parameters from a specific category, inference on the knowledge of that category can be performed.
>
> The data is filtered from open-source datasets, including The Pile, Skypile, RefinedWeb,  RedPajama and common crawl, as well as some synthetic data.Approximately 1 trillion tokens in total.
>
> # 6.Weaknesses6
>
> We have revised the introduction in the revised version.
>
> We hope our response addresses your concerns, but if there are any issues or areas we missed, please feel free to point them out.

---

> ### Author Response · Authors · 2024-11-26
>
> We have resolved all the issues in the weaknesses section and updated intro ,figure 2, and section  5.2 in the revised version.

---

### Official Review · Reviewer_ks1Z · 2024-11-04

**Soundness:** 2
**Presentation:** 1
**Contribution:** 2
**Rating:** 1
**Confidence:** 4

**Summary:**

This paper proposes a language model based on a router that selects domain-specific parameters. A base model parses an input, a router classifies the input into a category corresponding to a domain, then inference is done with parameters corresponding to the selected domain.

Each domain-specific parameter set is implemented using the last transformer layer of the base model replicated four times, and then trained with a "defined proportion" of domain-specific data and general data. The router has a similar parameterization, and is trained using classes corresponding to the domain-specific data.

The paper reports metrics on MMLU, C-Eval, GSM8K, and MATH, and also reports metrics from Llama2, Mistral, and Qwen1.5.

**Strengths:**

The idea of routing to domain-specific parameters is interesting (though more discussion of related work is needed, e.g. [1][2]).


[1] Branch-Train-Merge: Embarrassingly Parallel Training of Expert Language Models, Li et al 2022

[2] Branch-Train-MiX: Mixing Expert LLMs into a Mixture-of-Experts LLM, Sukhbaatar et al COLM 2024

**Weaknesses:**

The paper appears to be in an early stage. For example, key details of the method and experiment are not described or justified, and only 1 experiment (not fully described) has been performed. There is also missing discussion of key related work (e.g. [1], [2]). Here are some specific examples:

- The datasets have not been described. The data can substantially impact the downstream tasks that are evaluated in the experiment. Similarly, the number of tokens trained on is important and has not been reported.

- The evaluation is done on a proprietary evaluation pipeline, making reproducibility difficult.

- The experiments need a controlled comparison of the method against alternatives. Currently it is difficult to draw conclusions from the experiment provided. For example, IPLLM-24B and Qwen1.5-32B are not comparable since IPLLM has been trained on additional domain-specific data (which has not been specified, and may be relevant for the experimental comparison). One example comparison could be finetuning Qwen1.5-32B on the union of corpora that IPLLM finetunes on.

- Several claims made in the introduction and conclusion have not been justified. For example:
    - "Significant advantages in terms of reduced device memory requirements for both training and inference": this has not been justified. For example, the proposed method requires two forward passes at inference time, and an additional 4 x (number-of-domains + 1) layers to train.
    - "Enabling the model to learn new knowledge without catastrophic forgetting". This has not been justified experimentally.

- Ablations on key design decisions have not been done. For example, the routing strategy, number of layers, and the base model.

- Regarding novelty, Branch-Train-Merge [1] proposed to train different parts of the model independently on different subsets of the data (each subset corresponding to a domain, such as scientific or legal text). They also have a domain posterior that models the probability of a sequence belong to each domain (akin to the functionality of the proposed router). BTM and related follow-up work such as Branch-Train-MiX [2] should be discussed and compared with.

I would encourage the authors to continue improving the work since their idea has potential, but I believe the current manuscript is not yet ready for ICLR.

[1] Branch-Train-Merge: Embarrassingly Parallel Training of Expert Language Models, Li et al 2022
[2] Branch-Train-MiX: Mixing Expert LLMs into a Mixture-of-Experts LLM, Sukhbaatar et al COLM 2024

**Questions:**

Please address the points discussed in the Weaknesses.

---

### Official Review · Reviewer_e1e2 · 2024-11-10

**Soundness:** 1
**Presentation:** 1
**Contribution:** 1
**Rating:** 1
**Confidence:** 2

**Summary:**

This paper proposes to divide the LLMs into two step paradigm: (1) a routing step to have the transformer output the category of tasks; (2) for each category of tasks, a delegated transformer (base transformer + category special layers)  is applied to generate the outputs. Experiments are described in the paper to demonstrate the performance is comparable to existing models but not better.

**Strengths:**

• An interesting attempt to look for continuous scalable architecture for LLMs.

**Weaknesses:**

• It is not clearly to me how the proposal can achieve infinite-parameter models just by classifying the input to into different classes and training/using different networks for different classes.

**Questions:**

1. How the routing network (Equation 6) can be generalized to unseen categories so as to be generalized to infinite many categories? Is the routing token set fixed or open?
2. Is section 5 not completed in this version? It seems that a significant amount of texts are truncated between line 270 and line 282.

---

> ### Author Response · Authors · 2024-11-16
>
> Thank you very much for taking the time to review and provide comments. We are carefully examining the weaknesses and issues you pointed out, providing detailed explanations, and incorporating them into the revised version of the paper.
>
> # 1.Weaknesses:
>
> Traditional fixed-parameter models face significant challenges when additional parameters are introduced. The performance of tasks across all categories is affected, requiring full retraining for all tasks, which incurs prohibitive costs. Moreover, inference costs require loading all parameters into memory, making it infeasible for models with infinite parameters. Our IP-LLM model addresses these challenges by introducing a hierarchical and categorized structure. When adding parameters for a new category, only the parameters specific to that category need to be retrained, resulting in minimal training costs. For models with infinite parameters, inference only requires the base transformer, router, and category-specific layers, making the inference cost manageable.
>
> # 2.Questions1
>
>  The routing token set is open. When we need to generalize to unseen categories, we only need to add a set of parameters $f_{Y}$ and a routing token TokenY. Then, train this set of parameters with the new knowledge and retrain the router using the labels of both the new and old knowledge.
>
> \begin{equation}\label{key}
> 	R  = f^{new}_{router}(x^{\prime})
> \end{equation}
>
> When a large number of categories are added, if the routing accuracy decreases, it can be ensured by appropriately increasing the parameter capacity of $f_{router}$  and retraining it.
>
> We will incorporate this content into Section 3.
>
> # 3.Questions2
>
> Section 5 has already been completed. This was a formatting error, which we have corrected in the revised version of the paper.
>
>
>
>
> We hope our response addresses your concerns, but if there are any issues or areas we missed, please feel free to point them out.

---

> > ### Author Response · Authors · 2024-11-26
> >
> > We have resolved all the issues in the weaknesses section and updated section 3 and 5 in the revised version.

---

### Meta-Review · Area_Chair_p9Cg · 2024-12-08

**Metareview:**

This paper proposes a routed architecture for language models that selectively activates domain-specific parameters. While the core idea has merit, the work has significant limitations in evaluation methodology, empirical validation, and comparison to relevant baselines like Mixture-of-Experts models. The manuscript also requires substantial improvements in presentation and technical detail.

**Additional Comments On Reviewer Discussion:**

Reviewers were in consensus on the above assessment.

---

### Decision · Program_Chairs · 2025-01-22

Reject